# Higher versus Lower Oxygen Concentration during Respiratory Support in the Delivery Room in Extremely Preterm Infants: A Pilot Feasibility Study

**DOI:** 10.3390/children8110942

**Published:** 2021-10-20

**Authors:** Brenda Hiu Yan Law, Elizabeth Asztalos, Neil N. Finer, Maryna Yaskina, Maximo Vento, William Tarnow-Mordi, Prakesh S. Shah, Georg M. Schmölzer

**Affiliations:** 1Centre for the Studies of Asphyxia and Resuscitation, Neonatal Research Unit, Royal Alexandra Hospital, Alberta Health Services, Edmonton, AB T5H 3V9, Canada; blaw2@ualberta.ca; 2Division of Neonatology, Department of Pediatrics, University of Alberta, Edmonton, AB T6G 2R3, Canada; 3Sunnybrook Health Sciences Centre, Department of Paediatrics, University of Toronto, Toronto, ON M4N 3M5, Canada; elizabeth.asztalos@sunnybrook.ca; 4School of Medicine, University of California, San Diego, CA 92093, USA; nfiner@health.ucsd.edu; 5Sharp Mary Birch Hospital for Women and Newborns, San Diego, CA 92123, USA; 6Women and Children’s Health Research Institute, Department of Pediatrics, University of Alberta, Edmonton, AB T6G 1C9, Canada; yaskina@ualberta.ca; 7Health Research Centre, University and Polytechnic Hospital La Fe, 46026 Valencia, Spain; maximo.vento@uv.es; 8Division of Neonatology, University and Polytechnic Hospital La Fe, 46026 Valencia, Spain; 9Spanish Maternal and Infant Health and Development Network, Health Research Institute Carlos III, National Network, 46026 Madrid, Spain; 10NHMRC Clinical Trials Centre, University of Sydney, Camperdown, NSW 2050, Australia; wotarnowmordi@gmail.com; 11Department of Pediatrics, Mount Sinai Hospital, University of Toronto, Toronto, ON M5G 1X5, Canada; prakeshkumar.shah@sinaihealth.ca

**Keywords:** infant, newborn, delivery room, neonatal resuscitation, oxygen concentration

## Abstract

Background: Optimal starting oxygen concentration for delivery room resuscitation of extremely preterm infants (<29 weeks) remains unknown, with recommendations of 21–30% based on uncertain evidence. Individual patient randomized trials designed to answer this question have been hampered by poor enrolment. Hypothesis: It is feasible to compare 30% vs. 60% starting oxygen for delivery room resuscitation of extremely preterm infants using a change in local hospital policy and deferred consent approach. Study design: Prospective, single-center, feasibility study, with each starting oxygen concentration used for two months for all eligible infants. Population: Infants born at 23 + 0–28 + 6 weeks’ gestation who received delivery room resuscitation. Study interventions: Initial oxygen at 30% or 60%, increasing by 10–20% every minute for heart rate < 100 bpm, or increase to 100% for chest compressions. Primary outcome: Feasibility, defined by (i) achieving difference in cumulative supplied oxygen concentration between groups, and (ii) post-intervention rate consent >50%. Results: Thirty-four infants were born during a 4-month period; consent was obtained in 63%. Thirty (*n* = 12, 30% group; *n* = 18, 60% group) were analyzed, including limited data from eight who died or were transferred before parents could be approached. Median cumulative oxygen concentrations were significantly different between the two groups in the first 5 min. Conclusion: Randomized control trial of 30% or 60% oxygen at the initiation of resuscitation of extremely preterm neonates with deferred consent is feasible. Trial registration: Clinicaltrials.gov NCT03706586

## 1. Introduction

In the minutes following birth, normal oxygen saturations (SpO_2_) can be as low as 30% [1], which then increases to 85–95% over the next 7–10 min [2]. In term infants, resuscitation measures such as mask ventilation and supplemental oxygen may not be required to facilitate this transition. In contrast, most extremely preterm infants (<29 weeks’ gestation) will require respiratory support at birth [3]. In the past, preterm infants were resuscitated with 100% oxygen; however, in 2010, neonatal resuscitation guidelines recommended air or “less oxygen” as initial oxygen concentrations, which should be then be adjusted to meet age-dependent SpO_2_ targets [4]. Since then, have clinicians mostly transitioned to a lower starting oxygen concentration strategy in the resuscitation of preterm infants [5].

The most recent neonatal resuscitation guidelines published in 2020 recommend starting oxygen concentrations of 21–30%, based on uncertain evidence [6]. There is a lack of evidence for either overall benefit or harm in starting resuscitation with either lower (<30%) or higher (>65%) oxygen for preterm infants (i.e., <37 weeks’ gestation) [6,7]. Indeed, a recent survey of 630 clinicians from 25 countries showed that the majority would initiate preterm infant delivery room stabilization with 30–40% oxygen [5]. The balance is between the harms of hypoxia vs. hyperoxia. On one hand, hyperoxia may lead to the generation of oxygen free radicals, increased oxidative stress, and end-organ damage. Hyperoxia may also alter cerebral blood flow [8,9,10]. On the other hand, hypoxia may also result in harms such as brain injury and death [8,9,10]; failure to achieve oxygen saturation >80% at 5 min after birth has been associated with increased risk of IVH and death in both retrospective and prospective studies [11,12,13].

Randomized controlled trials and systematic reviews have evaluated different starting oxygen concentrations for resuscitation of preterm infants for the past 25 years, with varying definitions for “high” vs. “low” oxygen concentrations [7,13,14,15]. The conclusions for each trial and systematic review differ, highlighting the ongoing knowledge gap in this area. An individual patient meta-analysis by Oei et al. reported no difference in the overall risk of death with either lower (≤30%) or higher (≥60%) oxygen concentrations [14]. However, opposing results were seen in masked vs. unmasked trials, which the authors state could have represented a Type 1 error [14]. Furthermore, these meta-analyses also noted no differences in other common preterm morbidities when comparing low and high oxygen concentrations. [7,14] Finally, even with international cooperation, individual patient randomized controlled trials have had difficulty achieving target enrolment due to factors such as missed opportunities and clinicians declining to participate due to a perceived lack of equipoise [15,16].

Despite advances in perinatal and neonatal care, neonates remain susceptible to oxidative and deleterious effects from hyperoxia and hypoxia [8,9,10]. There is a need for large, multi-center international trials of sufficient sample size, using an alternate recruitment strategy, powered to look at both safety outcomes such as mortality and long-term outcomes such as neurodevelopment. In preparation for such a trial, to ensure that we can achieve a difference in supplied oxygen between the two intervention groups and that we can obtain an acceptable rate of enrolment, we performed an unblinded prospective, single-center feasibility study of 30% vs. 60% starting oxygen concentration at birth in extremely preterm infants to determine the feasibility of a multi-centered cluster-randomized crossover design using deferred consent.

## 2. Methods

This was a prospective, single-center, feasibility study comparing two starting oxygen concentrations (30% O_2_ vs. 60% O_2_) during initial respiratory support at birth. Between November 2018 and February 2019, all eligible infants born between 23 + 0 and 28 + 6 weeks’ gestation were included in the study. This feasibility study followed the design of a proposed cluster, crossover, randomized controlled trial with both interventions being implemented (Figure 1) using the CONSORT (Consolidated Standards of Reporting Trials) extension to randomized pilot and feasibility trials [17]. For this feasibility study, infants were managed at delivery with the first intervention (30% initial FiO_2_) for 2 months and then with the second intervention (60% initial FiO_2_) for another 2 months. This design mimicked a single-center participating in a cluster-randomized cross-over trial, where the local hospital policy would be changed to one of two randomized starting oxygen concentrations for a set recruitment period, with a similar pre-specified oxygen titration strategy for the entire trial duration. The study was carried out at the Royal Alexandra Hospital, Edmonton, a tertiary perinatal center admitting more than 350 infants with a birth weight of <1500 g to the neonatal intensive care unit (NICU) annually. The Royal Alexandra Hospital Research Committee and Health Ethics Research Board, University of Alberta (Pro00084090) approved the study and the study was registered at Clinicaltrials.gov (NCT03706586) [18].

### 2.1. Inclusion and Exclusion Criteria

Inborn infants between 23 + 0 and 28 + 6 weeks’ postmenstrual age were included in this study. Infants were excluded if they were (i) outborn (i.e., initial resuscitation not performed at the study center), (ii) born with a major congenital abnormality (e.g., congenital abnormalities that may affect oxygenation or neurodevelopmental outcomes), (iii) decision not to provide full resuscitation at birth, and (iv) if their parents declined to give consent after the study intervention.

### 2.2. Consent

We used a deferred consent approach, where written informed consent for use of patient data was sought from the parents as soon as possible after the birth after the initial resuscitation was completed. Further, our research ethics board approved limited data collection (i.e., delivery room data, in-hospital death, and major hospital morbidities) of infants for whom we did not have a chance to obtain parental consent due to death or being transferred to another facility within 72 h after birth. This approach allowed us to collect data for most infants receiving the intervention to allow the primary outcomes to be ascertained in all infants participating in the study.

### 2.3. Sample Size Calculation

For this pilot study, no sample size calculation has been performed. A convenient sample size of all infants within a 2-month time frame for each starting oxygen concentration group were recruited to determine the feasibility for recruitment for this intervention by using a cluster-randomized cross-over approach.

### 2.4. Blinding

Blinding was not feasible, as the first study intervention policy was assigned for two months and then switched to the alternate policy for another two months. However, the analysis team was blinded to group allocation.

### 2.5. Study Interventions

Delayed cord clamping for up to 60 s was attempted as per local hospital policy in all eligible infants. Other than starting oxygen concentration and oxygen titration strategy, all interventions such as mask ventilation, continuous positive airway pressure, intubation for poor respiratory effort or low heart rate, chest compressions, prevention of hypothermia by wrapping the infant in a polyethylene bag, and provision of appropriate medications were per the 2015 Neonatal Resuscitation Program guidelines. [7,8] The endotracheal intubation for the sole purpose of prophylactic surfactant administration was not allowed in the first 10 min after birth. After the first 10 min, ongoing SpO_2_ targeting and neonatal care were provided according to our center’s standard of care for both groups.

Infants remained in 30% or 60% O_2_ until 5 min of age unless the infant’s heart rate (HR) remained ≤100 beats per minute (bpm) and did not show a tendency towards progressive increase before reaching 5 min of age (oxygen concentration could then be increased by 10–20% every minute), or infant needs chest compression and/or epinephrine (oxygen concentration could then be increased to 100%) (Figure 1). No alterations in oxygen concentration were made for an infant who was responding to resuscitation efforts with HR progressively increasing. At 5 min of age, the clinical team assessed SpO_2_: If SpO_2_ was ≤85%, oxygen was increased by 10–20% every 60 s to achieve SpO_2_ of 90–95% at 10 min. If SpO_2_ was ≥95% oxygen was decreased (every 60 s) to maintain SpO_2_ of 90–95% at and beyond 10 min of age (Figure 1).

### 2.6. Outcomes

The primary outcome was the feasibility to perform a cluster trial by changing local hospital policy for starting oxygen concentration and oxygen titration strategy. Feasibility is defined by (a) ability to achieve difference between the two groups in supplied oxygen concentration during initial resuscitation, and (b) ability to obtain deferred consent >50% in infants who received the intervention. Secondary outcomes included: mortality prior to discharge from hospital, delivery room interventions (e.g., rate of intubation, rate of chest compression, use of epinephrine), mechanical ventilation, necrotizing enterocolitis, bronchopulmonary dysplasia (defined as oxygen and/or respiratory support at 36 weeks), retinopathy of prematurity, brain injury as indicated by abnormal cranial ultrasound. To quantify oxygen supplied over the first 5 and 10 min respectively, oxygen concentrations were added over the time period, representing a cumulative supplied oxygen concentration measure. For the first 5 min, the minimum cumulative supplied oxygen concentration would therefore be 21% × 5 min = 105, whereas the maximum total supplied oxygen would be 100% × 5 min = 500. Correspondingly, for the first 10 min, the minimum total supplied oxygen would be 21% × 10 min = 210, whereas the maximum total supplied oxygen would be 100% × 10 min = 1000.

## 3. Statistical Analysis

Data were analyzed as intention-to-treat and reported according to the CONSORT—Consolidated Standards of Reporting Trials extension to randomized pilot and feasibility trials (17). Data were compared using Student’s t-test for parametric and Mann-Whitney U test for nonparametric comparisons of continuous variables, and Fisher exact for categorical variables. The data are presented as mean (standard deviation (SD)) for normally distributed continuous variables and median (interquartile range (IQR)) when the distribution was skewed. *p*-values were 2-sided and *p* < 0.05 was considered statistically significant. Statistical analyses were performed with SPSS Statistics for Macintosh, Version 27.0 (IBM Corp, Armonk, NY, USA).

## 4. Results

A total of 34 infants (*n* = 14, 30% O_2_ group; *n* = 20, 60% O_2_ group) were born during the study period. None were excluded for major congenital anomalies or decision not to resuscitate at birth. Four infants (*n* = 2, 30% O_2_ group; *n* = 2, 60% O_2_ group) were excluded as parents declined consent after the study intervention had been performed. In the eight infants who died (*n* = 5) or were transferred to another hospital before parents could be approached for consent (*n* = 3), limited data were obtained (Figure 2). We achieved a consent rate of 63%. A total of 30 infants (*n* = 12, 30% O_2_ group; *n* = 18, 60% O_2_ group) were included in the final analysis (Figure 1). Demographics of included infants are presented in Table 1. The study protocol was followed in 10/12 infants (83%) and 13/18 infants (72%) in 30% O_2_ vs. 60% O_2_ groups, respectively. All protocol deviations related to increasing oxygen concentration more quickly than specified in the protocol. Despite protocol deviations, we achieved a difference in median cumulative supplied oxygen concentration between the two intervention groups for the first 5 min (240 (IQR 170-270) in the 30% O_2_ group vs. 315 (285–375) in the 60% O_2_ Group, *p* = 0.002) (Figure 3). However, the median total supplied oxygen concentration for the first 10 min was similar between groups (522 in the 30% O_2_ group vs. 561 in the 60% O_2_ Group, *p* = 0.172) Maximum oxygen concentration supplied in the first 10 min was also similar between groups (84% vs. 78%, *p* = 0.117).

In the 30% and 60% O_2_ groups, 12 (100%) and 17 (94%) received positive pressure ventilation (*p* = 1.000); and 6 (50%) and 6 (33%) were intubated, respectively (*p* = 0.458). One infant in the 30% O_2_ group received chest compression and epinephrine in the delivery room; this infant did not survive to admission to NICU. Infants in the 30% O_2_ group had trend towards lower SpO_2_ by 5 min of age (53% vs. 71%, *p* = 0.093) but similar mean heart rate (129 (29) vs. 119 (37) beats per minute, *p* = 0.47). Proportions of infants who had SpO_2_ <80% at 5 min were not statistically different (9/11 in the 30% O_2_ group vs. 9/18 in the 60% O_2_ group, *p* = 0.226) (Figure 4).

The number of infants diagnosed with bronchopulmonary dysplasia was 6 (67%) in the 30% O_2_ group vs. 1 (7%) in the 60% O_2_ group (*p* = 0.0049). No other differences in secondary neonatal outcomes were observed (Table 2).

## 5. Discussion

Recent neonatal guidelines recommend a starting oxygen concentration between 21% to 30%, based on very low-certainty evidence [6]. However, the optimal starting oxygen concentration for the resuscitation of extremely preterm infants remains unknown. Hyperoxia results in an increase in oxygen free radicals and decrease cerebral blood flow [8,9,10], while oxygen saturation of <80% at 5 min have been associated with increased mortality or neurodevelopmental disabilities [11,13].

A recent meta-analysis included 10 randomized trials and four cohort studies and demonstrated no significant risk or harm from either strategy [7]; however, included trials were small, with the largest trial being the To2rpido-trial with 287 patients [15]. In comparison, an individual patient meta-analysis analysis of eight trials (*n* = 768) reported that infants initially resuscitated with 21–30% vs. ≥60% O_2_ were less likely to achieve SpO_2_ ≥80%, which was associated with increased risk of major intraventricular hemorrhage and five times higher risk of death [14]. Unfortunately, most recent Individual patient randomized trials, including To2rpido and PRESOX (NCT01773746), have ceased early due to low enrolment [15,16].

Our study demonstrates the feasibility of using two starting oxygen concentrations for delivery room resuscitation of extremely preterm infants using deferred consent, mimicking single-center participation in a multi-center, cluster-randomized, crossover trial. We chose this design due to the often emergent nature of preterm births and the need to initiate immediate resuscitation in most situations. Consent was then obtained using a deferred consent model with written consent sought from the parents of these infants as soon as possible after birth to utilize data for research [19,20], as per the Canadian Tri-Council Policy Statement (TCPS) in Human Research guidelines. In Canada, TCPS policy explicitly sets out criteria allowing for “Exception to the requirement to seek prior consent”, which include: (i) necessity to answer the research question, (ii) lack of adverse impact on participants, (iii) justification of individual or society benefits compared with risks, (iv) minimal risk of interventions. In addition, this policy stipulates that the lack of prior consent “may be addressed through debriefing conducted as soon as possible following participants’ involvement in the research, and within a timeframe that allows participants to withdraw their data or biological materials” [21]. Additional criteria exist for altering the need for prior consent in emergency situations, where a potential participant requires immediate medical intervention, that there is no additional risk in the study intervention or even potential benefit, and that there is insufficient time to obtain consent from an authorized third party. These requirements for deferred consent differ between jurisdictions, which may complicate multi-national trials [20]. Using this approach, our consent rate was 63%; therefore, the estimated recruitment of over 100 subjects could be projected over a recruitment period of one year at a center of our size assuming a similar rate of preterm deliveries and consent rate. This might overcome limitations in recruitment encountered when using an antenatal consent approach, as experienced by previous randomized trials comparing higher vs. lower initial oxygen; in the To2rpido study, only 292 out of 6291 eligible infants were enrolled (4.6%) [15].

Another advantage to our approach is that infants born precipitously (where there is inadequate time for either antenatal consent or pre-delivery randomization) could be included, which could potentially increase the generalizability of results. An example of this was seen in the HIPSTER trial, where a change during the trial from prospective consent only to a mixed prospective/deferred consent strategy resulted in a larger proportion of eligible infants recruited and differences in infant demographics [22]. A mixed-methods study of parental opinions on deferred consent approach for a trial comparing delayed cord clamping vs. cord milking reported that this approach is highly acceptable for most families and might be preferable to prospective consent for some, specifically for interventions that are of low risk and within the standard of practice [23]. Parental perceptions of deferred consent are more mixed in trials examining interventions that are not considered low risk [23]. Our study interventions of using 30% vs. 60% O_2_ can be considered low risk and falls within the range of O_2_ used by clinicians in the course of neonatal resuscitation. A combined prospective/deferred consent strategy, where antenatal consent is obtained when possible, could be considered. However, this mixed consent strategy may be difficult to implement for a cluster-randomized trial, as all eligible infants born during the trial period would otherwise follow study interventions unless a family opts to not participate antenatally. Finally, in contrast to trials comparing 21% vs. 100% O_2_, a starting concentration of 30% vs. 60% O_2_ might also be more acceptable to clinicians [5].

Our difference in starting oxygen concentration was less than that of previous trials, yet we achieved a difference in supplied oxygen concentration over the first 5 min using a combination of (i) different initial oxygen concentrations and (ii) changing oxygen titration strategy based on a combination of heart rate and time-based saturation targets. The 30% O_2_ group had a lower mean SpO_2_ at 5 min compared with the 60% O_2_ group (53% vs. 71%, *p* = 0.093); however, this must be interpreted with caution as the 30% group had younger and smaller infants, lower rate of delayed cord clamping, and a lower proportion of female infants. Finally, both groups had a significant proportion of infants who did not achieve ≥80% SpO_2_ by 5 min, as oxygen was not adjusted before 5 min after birth. Therefore, future trials should consider earlier and/or more rapid oxygen concentration increases to achieve this goal, given the association between SpO_2_ <80% at 5 min with brain injury and poor neurological outcomes.

## 6. Limitations

We acknowledge that the number of subjects is too small to draw any conclusions regarding the relative efficacy of the two different oxygen concentrations. While not statistically significant, the 60% group had a higher mean birth weight and a higher proportion of female infants, which could explain the lower rates of bronchopulmonary dysplasia. We are currently organizing a multi-center cluster-randomized trial comparing 30% vs. 60% oxygen (HiLo-NCT02858583) to study this in a larger patient population [24].

## 7. Conclusions

Using a change in local hospital policy with deferred consent in a cross-over design can achieve a difference in supplied oxygen in the first 5 min of resuscitation with acceptable consent rate, making this study design feasible for a larger, multi-centered cluster-randomized crossover trial to study whether 30% vs. 60% would be the optimal starting oxygen for resuscitation of infants <29 weeks’ gestational age.

## Figures and Tables

**Figure 1 children-08-00942-f001:**
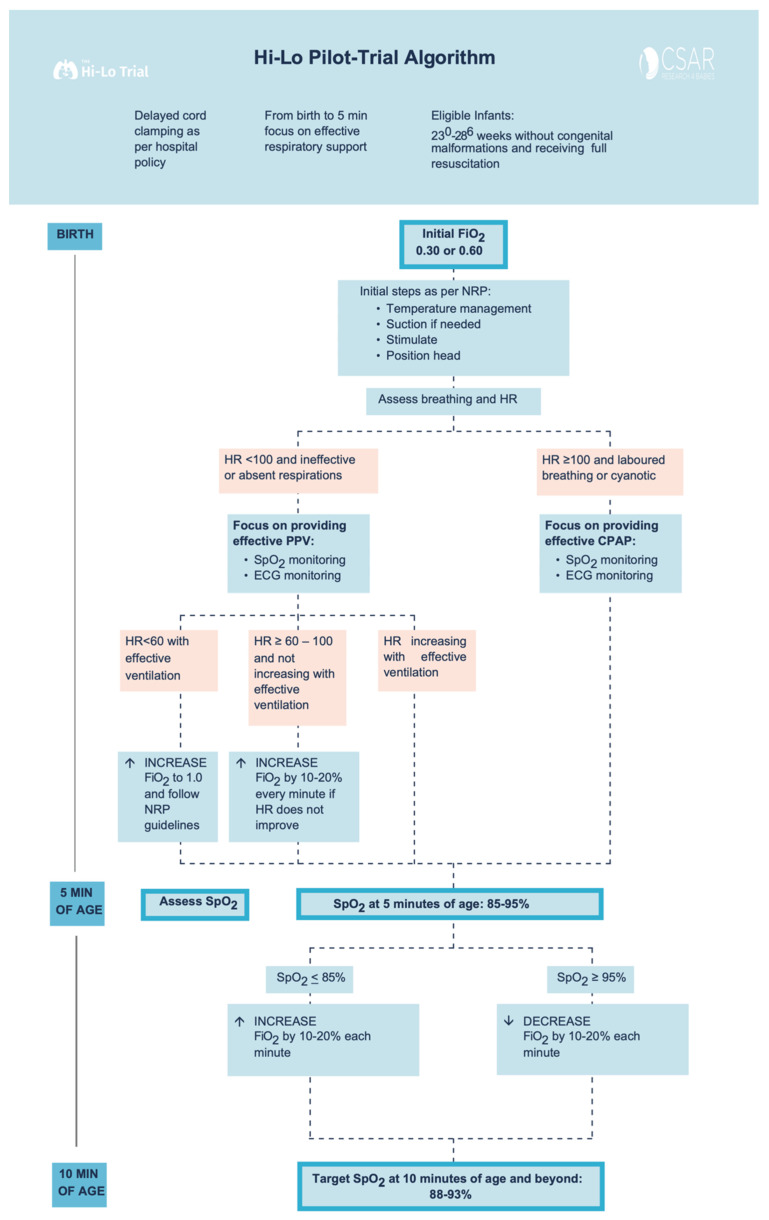
Study Interventions Flowchart.

**Figure 2 children-08-00942-f002:**
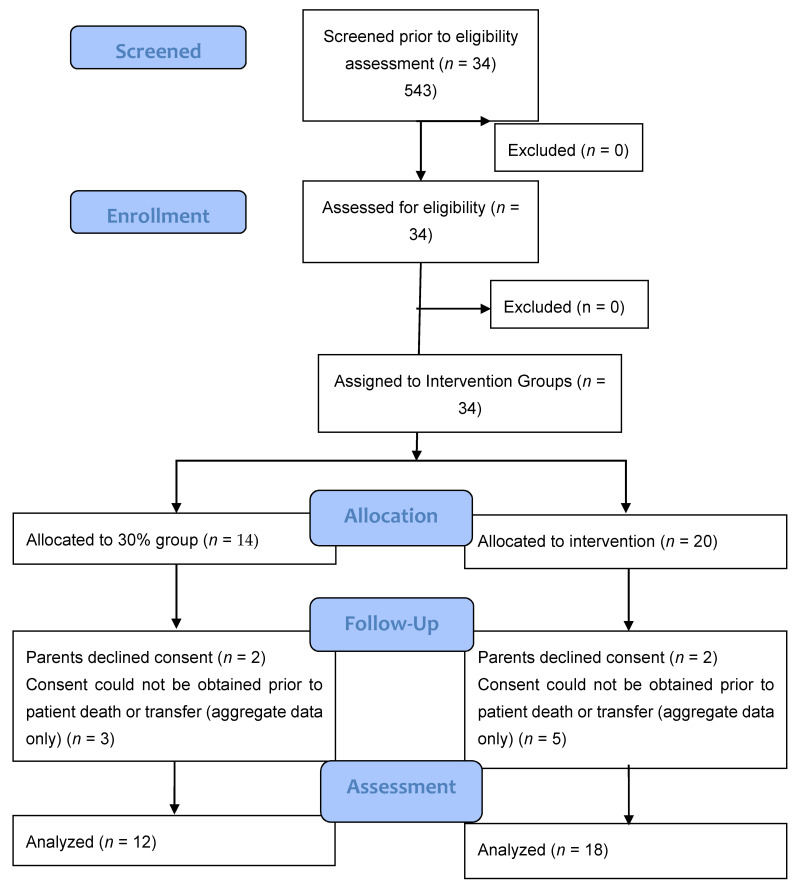
Consort–diagram.

**Figure 3 children-08-00942-f003:**
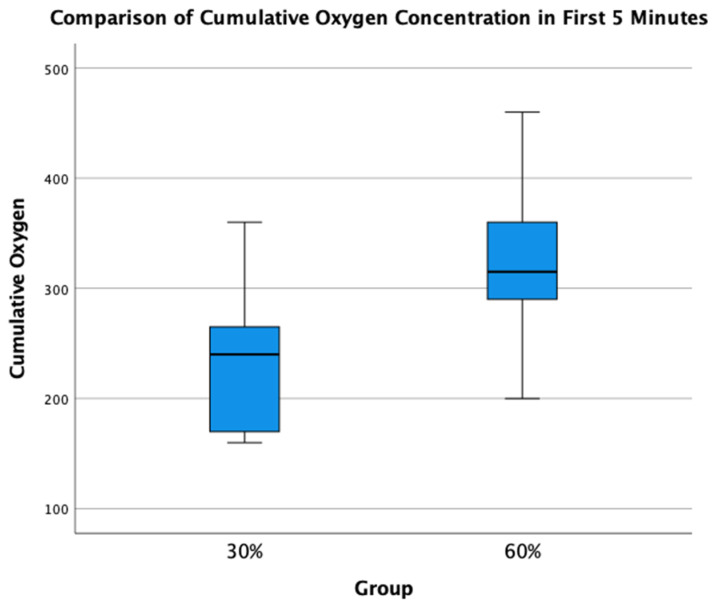
Comparison of Total Supplied Oxygen Concentrations for the first 5 min.

**Figure 4 children-08-00942-f004:**
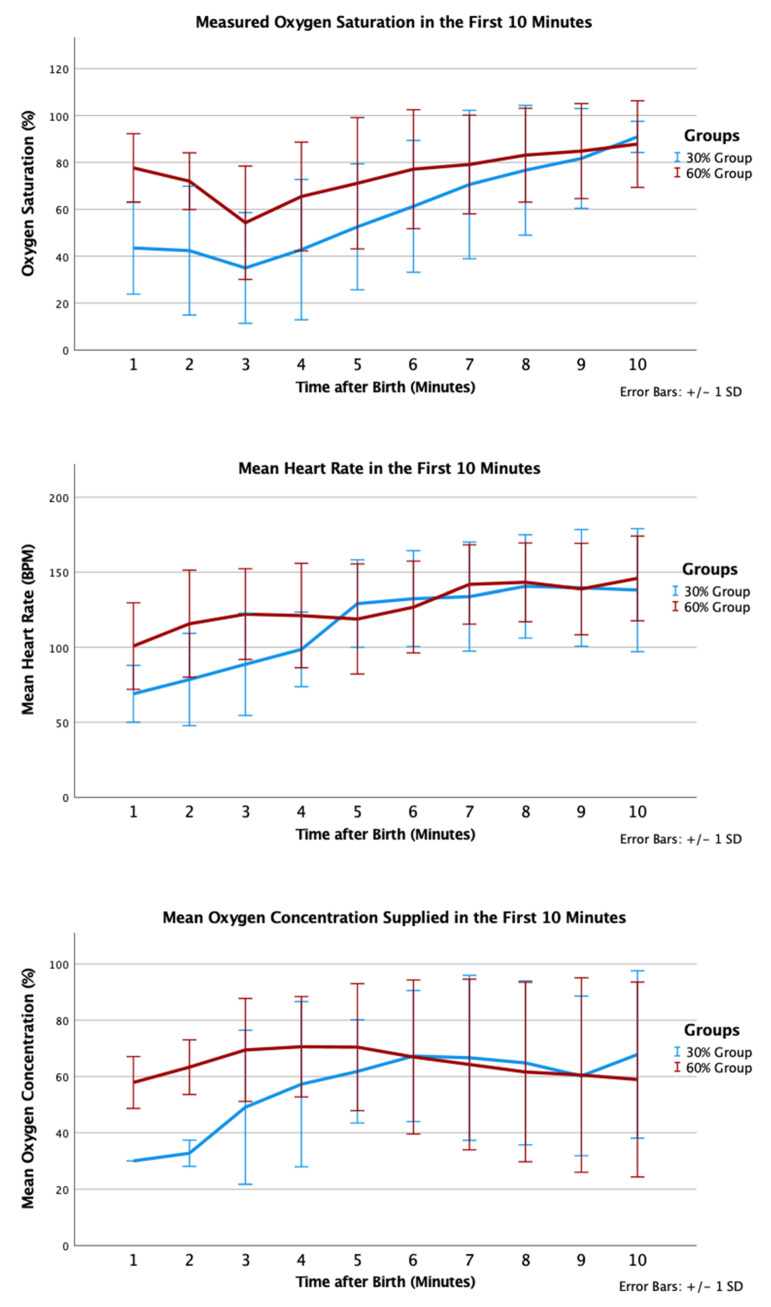
Mean oxygen saturation (%), oxygen saturation (%), and heart rate (bpm) in the first 10 min after birth according to group allocation.

**Table 1 children-08-00942-t001:** Demographics of study infants.

	30% Oxygen (*n* = 12)	60% Oxygen (*n* = 18)	*p*-Value
Birth weight (g)	847 (265)	1000 (247)	0.127
Gestational age (weeks)	25 (1.8)	26 (1.6)	0.131
Male (*n*) *	7 (58%)	6 (35%)	0.219
Antenatal steroids (*n*) *	11 (92%)	16 (94%)	1.00
Apgar 1 min ^#^	2 (1–5)	5 (2–5)	0.149
Apgar 5 min ^#^	6 (6–7)	7 (4–8)	0.258
Delayed Cord Clamping (*n*) *	5 (42%)	15 (83%)	0.045

Data are presented as mean (SD), unless indicated ^#^ median (IQR), * *n* (%).

**Table 2 children-08-00942-t002:** Secondary neonatal outcomes.

	30% Oxygen (*n* = 12)	60% Oxygen (*n* = 18)	*p*-Value
Surfactant	11 (92%)	13 (72%)	0.521
Death before discharge	3 (25%)	4 (22%)	1.000
Intraventricular hemorrhage all grades	7 (58%)	8 (44%)	0.701
Intraventricular hemorrhage grade ≥ 3	4 (33%)	4 (22%)	0.677
Patent ductus arteriosus	6 (50%)	9 (50%)	1.000
Necrotizing enterocolitis	2 (17%)	3 (17%)	1.000
Chronic lung disease in survivor	6 (67%)	1 (7%)	0.0049
Retinopathy of prematurity in survivor	3 (33%)	3 (27%)	0.643

Data are presented as *n* (%).

## Data Availability

All relevant data are in the manuscript.

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
