# Peer review of "Higher versus Lower Oxygen Concentration during Respiratory Support in the Delivery Room in Extremely Preterm Infants: A Pilot Feasibility Study"

_children, 2021, doi:10.3390/children8110942_

Round 1

Reviewer 1 Report

I have read this well-written paper about the feasibility of performing a cluster cross-over randomized trial with 2 initial oxygen concentration with great interest. The results of the study are straight forward and indicate good feasibility.

I do have some comments that I would like to share with the authors.

I am not completely sure why the authors decided that a feasibility study was urgently needed for the Hi-Lo trial. The authors describe that inclusion rate of pre-existing trials on high vs low initial O2 at birth appeared to be low, based on two issues.

  • The first is that they had difficulties with acquiring parental consent. While is commonly known that acquiring prospective parental consent in emergency situaties that is valid can be quite challenging. Therefore legislation and guidelines have been established, but they vary quite significantly between countries. Maybe the authors could elaborate in their discussion about the use of deferred consent and legislation about this in their country. Also, there have been concerns about obtaining deferred consent which I think the authors should discuss. In my experience, not all preterm deliveries are emergent, and maybe a combined approach of prospective and deferred consent would ethically be best. In addition, the authors state that "A deferred consent approach is highly acceptable for families and might be preferable to prospective consent". However, the paper that they refer to is mentioning conflicting results with regard to the perception of parents to deferred consent for DR studies. I would suggest that the authors elaborate their discussion about the use of deferred consent as a sole approach for obtaining valid parental consent.
  • Second, the authors mention that difficulties with obtaining a good inclusion rate was based on 'refusal of caregivers to participate' (Line 78). This last argument sounds negative in my opinion, as if they refused without reasons. Rather, in the trials that the authors refer to, it was mentioned that "recruitment difficulties arose due to the lack of equipoise"(Oei et al. Pediatrics 2017). Maybe the authors could change this section of the introduction?

The chart that describes the intervention protocol (with regard to oxygen titration) is not readable, it includes symbols instead of letters. So the design of the study cannot objectively be checked.

Have the authors also evaluated the time that SpO2 values fell within/above/below target ranges? In their titration protocol it was included that if heart rate is > 100 bpm, FiO2 will only be titrated after 5 min, resulting in an increased risk of hypoxemia and hyperoxemia in this first period.

My other major comment is about the conclusion. The authors mainly discuss findings about SpO2 values, which are not their primary outcome and these results have not been presented in the results section (e.g. timepoint at which a SpO2 of >80% was achieved). Based on the results presented and the study design, only valid conclusions can be made based on feasibility (i; ability to achieve a difference in supplied oxygen concentration, and ii; ability to obtain deferred consent in >50% of infants).

I do have some minor comments:

  • In the abstract, the hypothesis presented is not a hypothesis but more an aim.
  • Line 65: add "at 5 min after birth"
  • Line 106: Ciinicaltrials.gov should be changed in Clinicaltrials.gov
  • The authors state that it is feasible to include 200 infants in a year, based on their admission rate of 350 infants <1500 grams a year. However, in the study period of 4 months only 34 infants have been born <29 weeks, which (if this would continue for a year) would give you 102 infants a year..
  • In the flowchart in the results section, the number of infants screened should be checked.
  • In the conclusion it is discussed why infants in the 60% group could have had a higher SpO2 at 5 min. However, the authors did not include there the incidence of delayed cord clamping, which was significantly higher in the 60% group. This could have influenced oxygenation as well.
  • Line 238: do the authors mean by mask ventilation only PPV? Or CPAP also?
  • For all results: if mean values are presented with a p-value, add SD.

Author Response

Thank you for your review of our manuscript.  Please see the follow for our response.  We hope that we have adequately addressed your thoughtful and helpful comments and questions.

I have read this well-written paper about the feasibility of performing a cluster cross-over randomized trial with 2 initial oxygen concentration with great interest. The results of the study are straight forward and indicate good feasibility.

  • Thank you.

I do have some comments that I would like to share with the authors.

I am not completely sure why the authors decided that a feasibility study was urgently needed for the Hi-Lo trial.

  • To clarify, we have changed the final sentence in the introduction to clearly specify why a feasibility study was needed: “In preparation for such a trial, to ensure that we can achieve a difference in supplied oxygen between the two intervention groups and that we can obtain an acceptable rate of enrolment, we performed an unblinded prospective, single-center feasibility study of 30% vs. 60% starting oxygen concentration at birth in extremely preterm infants to determine the feasibility of a multi-centered cluster-randomized crossover design using deferred consent.”

The authors describe that inclusion rate of pre-existing trials on high vs low initial O2 at birth appeared to be low, based on two issues.

  • The first is that they had difficulties with acquiring parental consent. While is commonly known that acquiring prospective parental consent in emergency situations that is valid can be quite challenging. Therefore, legislation and guidelines have been established, but they vary quite significantly between countries. Maybe the authors could elaborate in their discussion about the use of deferred consent and legislation about this in their country.
  • Thank you. Indeed, legislation and policies surrounding deferred consent differs between jurisdictions and can therefore be challenging for a multi-centered international trial to navigate. We have elaborated on this topic, with discussion of the criteria in Canada for use of deferred consent.  The discussion for this now reads:

    “Consent was then obtained using a deferred consent model with written consent sought from the parents of these infants as soon as possible after birth to utilize data for research(19,20), as per the Canadian Tri-Council Policy Statement (TCPS) in Human Research guidelines. In Canada, TCPS policy explicitly sets out criteria allowing for “Exception to the requirement to seek prior consent”, which include: i) necessity to answer the research question, ii) lack of adverse impact on participants, iii) justification of individual or society benefits compared with risks, iv) minimal risk of interventions. In addition, this policy stipulates that the lack of prior consent “may be addressed through debriefing conducted as soon as possible following participants’ involvement in the research, and within a timeframe that allows participants to withdraw their data or biological materials.”(21)  Additional criteria exist for altering the need for prior consent in emergency situations, where a potential participant requires immediate medical intervention, that there is no additional risk in the study intervention or even potential benefit, and that there is insufficient time to obtain consent from an authorized third party. These requirements for deferred consent differ between jurisdictions, which may complicate multi-national trials.(20)”

Also, there have been concerns about obtaining deferred consent which I think the authors should discuss. In my experience, not all preterm deliveries are emergent, and maybe a combined approach of prospective and deferred consent would ethically be best. In addition, the authors state that "A deferred consent approach is highly acceptable for families and might be preferable to prospective consent". However, the paper that they refer to is mentioning conflicting results regarding the perception of parents to deferred consent for DR studies. I would suggest that the authors elaborate their discussion about the use of deferred consent as a sole approach for obtaining valid parental consent.

  • Thank you. For some study designs, a mixed approach may be possible. This would particularly be true for individual patient randomized trials – thus for situations where there is time for consent discussion families can opt in or out of a study, but in emergent situations we could use deferred consent. For a cluster-randomize cross-over design, mixed consent may be very difficult to implement. As in our study, local hospital policy was changed during the study period to reflect study interventions. For a family to decline participation antenatally would then necessitate deviation from current hospital policy for that case only, which would be difficult to operationalize.   This discussion now reads as follows:

    “Another advantage to our approach is that infants born precipitously (where there is inadequate time for either antenatal consent or pre-delivery randomization) could be included, which could potentially increase the generalizability of results. An example of this was seen in the HIPSTER trial, where a change during the trial from prospective consent only to a mixed prospective / deferred consent strategy resulted in a larger proportion of eligible infants recruited and differences in infant demographics.(22)  A mixed methods study of parental opinions on deferred consent approach for a trial comparing delayed cord clamping vs. cord milking reported that this approach is highly acceptable for most families and might be preferable to prospective consent for some, specifically for interventions that are of low risk and within standard of practice.(23) Parental perceptions of deferred consent is more mixed in trials examining interventions that are not considered low risk.(23) Our study interventions of using 30% vs 60% O2 can be considered low risk and falls within the range of O2 used by clinicians in the course of neonatal resuscitation.  A combined prospective / deferred consent strategy, where antenatal consent is obtained when possible, could be considered. However, this mixed consent strategy may be difficult to implement for a cluster-randomized trial, as all eligible infants born during the trial period would otherwise follow study interventions unless a family opts to not participate antenatally. Finally, in contrast to trials comparing 21% vs 100% O2, a starting concentration of 30% vs. 60% O2 might also be more acceptable to clinicians.(5)”
  • Second, the authors mention that difficulties with obtaining a good inclusion rate was based on 'refusal of caregivers to participate' (Line 78). This last argument sounds negative in my opinion, as if they refused without reasons. Rather, in the trials that the authors refer to, it was mentioned that "recruitment difficulties arose due to the lack of equipoise"(Oei et al. Pediatrics 2017). Maybe the authors could change this section of the introduction? 
    • Yes, in Oei et al. “Clinicians refuse to participate” was listed as a frequent exclusion, however, this is perhaps better worded as you suggest.  We have changed this sentence to read: “Finally, even with international cooperation, individual patient randomized controlled trials have had difficulty achieving target enrolment due to factors such as missed opportunities and clinicians’ declining to participate due to a perceived of lack of equipoise.”

The chart that describes the intervention protocol (with regard to oxygen titration) is not readable, it includes symbols instead of letters. So the design of the study cannot objectively be checked.

  • We apologize. A corrected version has been uploaded.

Have the authors also evaluated the time that SpO2 values fell within/above/below target ranges? In their titration protocol it was included that if heart rate is > 100 bpm, FiO2 will only be titrated after 5 min, resulting in an increased risk of hypoxemia and hyperoxemia in this first period.

  • Unfortunately, there were some missing values for the first 1-3 minutes for SpO2 for this comparison to be accurate. Thus, we utilized SpO2 at 5 minutes as a marker of hypoxemia instead.  For interest, more infants in the 60% group had SpO2>85% at 5 minutes (41% vs 18%, p=0.45), but this was not statistically significant.

My other major comment is about the conclusion. The authors mainly discuss findings about SpO2 values, which are not their primary outcome and these results have not been presented in the results section (e.g. timepoint at which a SpO2 of >80% was achieved). Based on the results presented and the study design, only valid conclusions can be made based on feasibility (i; ability to achieve a difference in supplied oxygen concentration, and ii; ability to obtain deferred consent in >50% of infants).

  • Thank you. We have adjusted the conclusion to focus on the feasibility of the study. It now reads “Using a change in local hospital policy with deferred consent in a cross-over design can achieve difference in supplied oxygen in the first 5 minutes of resuscitation with acceptable consent rate, making this study design feasible for a larger, multi-centered cluster-randomized crossover trial to study whether 30% vs. 60% would be the optimal starting oxygen for resuscitation of infants <29 weeks gestational age.”

I do have some minor comments:

  • In the abstract, the hypothesis presented is not a hypothesis but more an aim.
    • This has been corrected to read: “Hypothesis: It is feasible to compare 30% vs 60% starting oxygen for delivery room resuscitation of extremely preterm infants using a change in local hospital policy and deferred consent approach.”

  • Line 65: add "at 5 min after birth"
    • This has been corrected. It now reads: “…failure to achieve oxygen saturation >80% at 5 min after birth have been associated with increased risk of IVH and death in both retrospective and prospective studies.”

  • Line 106: Ciinicaltrials.gov should be changed in Clinicaltrials.gov
    • Corrected

  • The authors state that it is feasible to include 200 infants in a year, based on their admission rate of 350 infants <1500 grams a year. However, in the study period of 4 months only 34 infants have been born <29 weeks, which (if this would continue for a year) would give you 102 infants a year.
    • We have changed this to read: “Our consent rate was 63%; therefore, an estimated recruitment of over 100 subjects could be projected over a recruitment period of one year at a center of our size assuming a similar rate of preterm deliveries and consent rate.”

  • In the flowchart in the results section, the number of infants screened should be checked.
    • Thank you.  During the study time period, there were no infants who were delivered who did not receive resuscitation (planned palliative care) or who had major congenital anomalies for which our resuscitation teams would exclude automatically from our resuscitation studies, and thus the screened number is the same as eligibility assessment.

  • In the conclusion it is discussed why infants in the 60% group could have had a higher SpO2 at 5 min. However, the authors did not include there the incidence of delayed cord clamping, which was significantly higher in the 60% group. This could have influenced oxygenation as well.
    • We have changed the conclusion as previously discussed, but also added mention of delayed cord clamping as a factor in the discussion as follows: “The 30% O2 group had a lower mean SpO2 at 5 min compared with the 60% O2 group (53% vs. 71%, p=0.093); however, this must be interpreted with caution as the 30% group had younger and smaller infants, lower rate of delayed cord clamping, and a lower proportion of female infants.”

  • Line 238: do the authors mean by mask ventilation only PPV? Or CPAP also?
    • Mask ventilation here is PPV. This is clarified: “In the 30% and 60% O2 groups, 12 (100%) and 17 (94%) received positive pressure ventilation (p=1.000); and 6 (50%) and 6 (33%) were intubated, respectively (p=0.458).”

  • For all results: if mean values are presented with a p-value, add SD.
    • SDs are already listed in Tables 1 where appropriate. We have added IQRs and SD to a few results not listed in the Tables that are presented in the results section.

Thank you again for your thoughtful review.

Reviewer 2 Report

Important single centre study comparing FiO2 0.3 vs. 0.6 in extremely preterm neonates resuscitated in the delivery room. It is concluded, that resuscitation with FiO2 0.3 vs. 0.6 is possible in the delivery room 

Figure 1 is not readable in the version available to this reviewer

Limitations: no control group with room air FiO2 0.21 has been included in this study

Correct few typos

Author Response

(The authors gave the same response as above.)

Round 2

Reviewer 1 Report

I would like to thank the authors for their responses.